# Nano-Formulations of Natural Antioxidants for the Treatment of Liver Cancer

**DOI:** 10.3390/biom14081031

**Published:** 2024-08-19

**Authors:** Mariateresa Cristani, Andrea Citarella, Federica Carnamucio, Nicola Micale

**Affiliations:** 1Department of Chemical, Biological, Pharmaceutical and Environmental Sciences, University of Messina, Viale Ferdinando Stagno D’Alcontres 31, I-98166 Messina, Italy; mariateresa.cristani@unime.it; 2Department of Chemistry, University of Milan, Via Golgi 19, I-20133 Milano, Italy; andrea.citarella@unimi.it; 3Center of Pharmaceutical Engineering and Sciences, Department of Pharmaceutics, Virginia Commonwealth University, Richmond, VA 23284, USA

**Keywords:** antioxidants, liver cancer, nano-formulations, drug delivery, natural compounds, hepatocellular carcinoma, oxidative stress

## Abstract

Oxidative stress is a key factor in the pathological processes that trigger various chronic liver diseases, and significantly contributes to the development of hepatocarcinogenesis. Natural antioxidants reduce oxidative stress by neutralizing free radicals and play a crucial role in the treatment of free-radical-induced liver diseases. However, their efficacy is often limited by poor bioavailability and metabolic stability. To address these limitations, recent advances have focused on developing nano-drug delivery systems that protect them from degradation and enhance their therapeutic potential. Among the several critical benefits, they showed to be able to improve bioavailability and targeted delivery, thereby reducing off-target effects by specifically directing the antioxidant to the liver tumor site. Moreover, these nanosystems led to sustained release, prolonging the therapeutic effect over time. Some of them also exhibited synergistic effects when combined with other therapeutic agents, allowing for improved overall efficacy. This review aims to discuss recent scientific advances in nano-formulations containing natural antioxidant molecules, highlighting their potential as promising therapeutic approaches for the treatment of liver cancer. The novelty of this review lies in its comprehensive focus on the latest developments in nano-formulations of natural antioxidants for the treatment of liver cancer.

## 1. Introduction

Liver cancer is a life-threatening illness and is one of the most aggressive and steadily increasing forms of cancer in the world [1]. Recent statistics indicate that it is the sixth-most common form of cancer and affects more men (fifth-most common form of cancer) than women (ninth-most common form of cancer) [2]. However, these findings refer to primary liver cancer, also known as primary hepatic cancer or primary hepatic malignancy, which starts and develops in the liver parenchyma or other structures within the liver, such as the bile duct, blood vessels, and immune cells [3,4]. In contrast, most liver cancers are secondary or metastatic, meaning they originate in other parts of the body (e.g., colon, breast, or lungs) and then spread to the liver [5]. Among the sub-types of primary liver cancers, hepatocellular carcinoma (HCC), which forms from hepatocytes, is the most common (80–90%) [6,7], followed by cancer of the bile duct, namely intrahepatic cholangiocarcinoma (~6%), angiosarcoma and hemangiosarcoma (0.1–2.0%), two rare and very aggressive forms of cancer that originate from the endothelial layer of blood vessels and metastasize rapidly [8], and hepatoblastoma (~1%), a type of cancer that develops in immature liver cells and primarily affects children under the age of 15 years [9]. Risk factors that increase the incidence of primary liver cancer include, in order, cirrhosis due to chronic viral infections (such as hepatitis B and hepatitis C) or excessive alcohol consumption, metabolic diseases (such as nonalcoholic steatohepatitis, nonalcoholic fatty liver diseases, obesity, and diabetes) [10], hereditary liver diseases (such as hemochromatosis and Wilson’s disease) [11,12], exposure to aflatoxins [13], iron accumulation [14], and certain biliary tract diseases (e.g., fascioliasis, a zoonotic infectious disease caused by parasitic worms of the genus *Fasciola*) [15].

More and more studies have recently demonstrated a clear correlation between oxidative stress (OS) and the onset/development of liver cancer [16,17,18,19,20,21,22,23]. OS is defined as a non-physiological condition of the body in which there is a marked imbalance between the production of reactive species with oxidant properties and the functioning of endogenous defense systems with antioxidant capacity. These reactive species are generally classified into ROS (reactive oxygen species), RNS (reactive nitrogen species), RSS (reactive sulfur species), and RCS (reactive chlorine species). ROS are those produced the most and are by far the most harmful. They include superoxide anion radical (O_2_•^−^), hydroxyl radical (OH•), hydroxyl ion (OH−), hydrogen peroxide (H_2_O_2_), singlet oxygen (^1^O_2_), and ozone (O_3_) [24]. They can be produced by exogenous sources (e.g., UV radiation, toxic substances, drugs), physiological changes (e.g., stress conditions, chronic inflammatory states, aging), or endogenous sources (e.g., byproducts of the mitochondrial respiratory chain, transmembrane enzymes of the NADPH oxidase family, myeloperoxidase in phagocytes) [25]. Under normal physiological conditions, OS is kept under control by the adequate presence of metabolic pathway cofactors with free radical scavenger properties (e.g., vit. C, vit. E, CoQ10) and by the activity of antioxidant enzymes such as catalase (CAT), superoxide dismutase (SOD), and glutathione reductase (GRS) [26]. In contrast, uncontrolled OS plays a pivotal role in carcinogenesis and liver cell progression; its action occurs through multiple pathways, including direct oxidative damage to DNA, instability of genetic material, interference with signaling pathways of tumor growth and programmed cell death, production of cytokines, chemokines and mediators of the immune system, activation of pathways involved in metastasis, dysfunction of cytoplasmic organelles, angiogenesis, and increased drug resistance [27]. The mechanisms by which OS intervenes in the onset and progression of liver cancer are depicted and briefly described in Figure 1.

In view of all the evidence demonstrating the involvement of OS in the development of liver cancer and given the latter’s crucial role as the primary organ of detoxification and regulator of homeostasis in the human body, the use of natural antioxidants is becoming increasingly important both in terms of prevention and therapeutic treatment of this disease [28,29,30,31]. The plant world offers the greatest resources in this regard, with its abundance of (poly)phenolic substances whose beneficial effects on health have been known since ancient times and whose use as medicinal herbs is part of traditional medicine in many countries [32]. Unfortunately, large-scale use of these compounds as drugs has not been very successful so far because of their poor oral bioavailability, mainly due to their poor solubility in water and rapid first-pass metabolism [33]. In addition, one must consider a certain chemical instability of these compounds, which are easily oxidized in air and light [34].

In this context, the use of nanotechnology offers a wide range of versatile and more effective solutions than older, nonspecific cancer therapies [35,36,37,38]. Nanoparticles (NPs), particles typically ranging in size from 1 to 200 nm (although slightly larger sizes are universally accepted for applications in nanomedicine), with their large surface area can greatly increase the solubility of natural antioxidants (entrapped in their core or chemically bound or adsorbed on the surface), protect them from biodegradation or physicochemical degradation, promote their passage through biological barriers, provide sustained and gradual release, and offer stimuli-responsive delivery systems [39]. In addition, the surface area of NPs, depending on the material used for their formation, can be exploited for the construction of tumor-specific delivery vehicles by linking to it (mostly via covalent bonding) targeting agents, thus limiting as much as possible off-target effects. Nonetheless, NPs’ surface area can be further functionalized/exploited to obtain drug delivery systems for combination therapy by joining natural antioxidants with standard anticancer drugs, reducing dosage and side effects of the latter due to synergistic effects [40,41]. 

Several types of nanostructured materials have been intensively studied over the years for the possible treatment of liver cancer (more specifically for the treatment of HCC): NPs containing natural polymers as the basic component (e.g., chitosan, dextran); NPs constructed from synthetic polymers and/or copolymers (e.g., poly(ε-caprolactone), poly(α-hydroxybutyric acid), poly(lactide-*co*-glycolide)); self-assembled nanostructures (e.g., liposomes, dendrimers, micelles); NPs obtained from biomacromolecules (e.g., albumin) [42]. The antioxidant profile of natural molecules has been improved mainly by techniques of loading onto the surface of nanostructures, entrapping in nanogels or in the core and/or core-shell spaces of hollow nanospheres [43,44,45,46]. In terms of biocompatibility and toxicity, carbon nanotubes, cationic NPs, metallic NPs, and silica-based NPs have been less successful for potential use against HCC. Carbon nanotubes have been associated with changes in liver cell morphology, platelet aggregation, toxicity due to the release of carbon residues, and, most importantly, the development of OS that could further exacerbate the pathological state of the liver [47,48]. Cationic NPs essentially cause blood clotting and hemolysis [49]. Metallic NPs, particularly silver NPs, have also been associated with the development of OS with massive ROS production [50,51]. As for silica NPs, in addition to OS, several studies have shown increased biomarkers of liver damage (e.g., alanine aminotransferase (ALT), aspartate aminotransferase (AST), alkaline phosphatase (ALP)) and other hepatotoxic effects [52,53,54]. 

This review focuses on the growing intersection of nanotechnology and natural antioxidants in the treatment of liver cancer, a field that has made rapid progress but has yet to be fully explored in comprehensive reviews. Moreover, considering the increasing global liver cancer burden and the urgent need for more effective, targeted, and less toxic therapies, the timing of this review is particularly crucial. Its novelty lies in its detailed examination of the past decade’s nano-formulations specifically designed to enhance the therapeutic potential of natural antioxidants, offering insights into how these innovations are beginning to overcome the limitations of traditional liver cancer treatments. 

## 2. Nano-Formulations Containing (Poly)phenolic Antioxidants

Plant-derived (poly)phenols are universally recognized by the scientific community as major OS effectors and are currently the subject of extensive research in the field of oncology [55,56]. Based on the chemical structure, these phytochemicals are classified into phenolic acids, stilbenes, flavonoids, coumarins, and lignans [57]. Among these, resveratrol (RES), a stilbene-type aflatoxin, has gained greater attention by virtue of its wide distribution in the plant kingdom (e.g., grapes, berries, legumes, peanuts, and various grasses) and, more importantly, because of its broad spectrum of pharmacological activities, which include (in addition to antioxidant) anti-inflammatory, anticarcinogenic, antiplatelet, cardioprotective, immunomodulatory, hyperlipidemic, neuroprotective, and vasorelaxant activities [58,59,60]. Naturally occurring RES (3,5,4′-trihydroxystilbene; Figure 2A) can be found as *cis*- or *trans*-isomer, with the latter being the more stable, abundant, and therapeutically active of the two. Like all polyphenols, RES suffers from physicochemical instability and pharmacokinetic issues. Although it is well absorbed by the intestinal mucosa (~75%), RES is extensively metabolized both at the intestine and liver levels to form inactive 3-*O*-glucuronide and 3-*O*-sulfate derivatives. In addition, RES binds to plasma proteins and low-density lipoproteins (LDLs), resulting in only small amounts of free RES in the systemic circulation. As a result, RES has a very short plasma half-life (8–14 min) [61].

Under this scenario, nanotechnology has invested considerable resources and made efforts in the attempt to address problems [62,63]. Lian B. and co-workers were able to increase ~6-fold the bioavailability of RES (assessed in Sprague-Dawley rats) after intravenous administration of an intriguing nano-formulation composed of folic acid (FA) and human serum albumin (HSA), in which RES was encapsulated by a high-pressure fluid emulsification method (Figure 2) [64]. FA, employed as an imaging and cancer-targeting ligand because HCC cells (HepG2) overexpress FA receptors on their surfaces, was first selectively activated at its terminal free -COOH group as an *N*-hydroxysuccinimide (NHS) ester moiety (DCC as a coupling reagent) and then conjugate to HSA under alkaline conditions (TEA in DMSO). The resulting FA-HSA conjugate (aqueous solution) was then mixed with RES (organic solution) and the resulting emulsion was homogenized by means of a high-pressure nanometer high-efficiency device. The final FA-HSA-RES-NPs showed spherical shape, proper size (~102 nm), high encapsulation efficiency (EE; ~98%, with RES amorphously encapsulated and in low crystallinity) and drug loading (DL; ~15%), superior cell uptake rate and antiproliferative activity in vitro than RES (IC_50_ = 110.8 µM vs. IC_50_ = 152.7 µM in HepG2 cells), and optimal tumor-targeting ability (detected in vivo by near-infrared imaging in tumor-bearing mice). In vitro drug release (DR) was claimed to be slow and continuous [64].

The same authors previously developed a composed nano-formulation in which RES was encapsulated in a glycyrrhizic acid(GA)-HSA conjugate by means of a similar high-pressure homogenization emulsification method [65]. In that case, GA was employed as a targeting ligand instead of FA because GA receptors are highly present on the cell surface of hepatocytes, wherein GA accumulates and toward which it exerts hepatoprotective and detoxifying effects [66]. Indeed, several nanoscale drug delivery systems containing GA for the treatment of HCC have been the focus of recent research works or literature reviews [67,68,69]. Analogous to the work discussed above, the free -COOH groups of the targeting ligand were exploited for conjugation with HSA under alkaline conditions and the resulting GA-HSA conjugates were loaded with RES as previously described. GA-HSA-RES-NPs were nearly spherical, ~108 nm in size, and had EE and DL of 83.6% and 11.5%, respectively. RES was released from the nanosystem slowly and continuously. The impact on the antiproliferative activity against HepG2 cells was slightly superior (1.5-fold) compared to the previous nano-formulation, i.e., GA-HSA-RES-NPs → IC_50_ = 62.5 µg/mL vs. RES → IC_50_ = 95.5 µg/mL. In vitro cell uptake and in vivo body distribution studies (near-infrared fluorescence imaging method in H22 tumor-bearing mice) confirmed the validity of this nano-formulation [65]. 

Noteworthy improvements in RES in terms of biological activity profile have also been achieved by Zhang D. et al. with the development of RES-loaded gold-based NPs (RES-AuNPs) [70]. AuNPs are quite exploited in cancer therapy as nanocarriers due to their unique properties, including remarkable biocompatibility, low toxicity, high surface-to-volume ratio, and functional versatility (e.g., tunable surface plasmon resonance that can be exploited for localized photothermal therapy and ease of synthesis for eco-friendly chemistry applications) [71,72]. RES in this case was used both as a green reducing agent for the starting Au salt (i.e., chloroauric acid; HAuCl_4_) and bioactive component of the nano-formulation. RES-AuNPs showed higher antiproliferative activity than free RES (IC_50_ = 3.84 µg/mL vs. IC_50_ = 24.74 µg/mL in HepG2 cells) and lower toxicity against normal cells (IC_50_ = 66.7 µg/mL vs. IC_50_ = 8.3 µg/mL in L02 cells). Further in vitro studies performed on HepG2 cells showed that this nano-formulation has stronger effects than free RES on inducing apoptosis via downregulation of pro-caspase-9, pro-caspase-8, PI3K, and Akt, and upregulation of caspase-8 and Bax, thus suggesting that the process might be mediated both via dysfunction of the mitochondrial-related pathway (the one mainly involved according to the uptake studies) and the PI3K/Akt signaling pathway. Furthermore, in vivo studies (xenograft murine model) indicated that RES-AuNPs remarkably suppress tumor growth and angiogenesis and confirmed the induction of apoptosis [70]. 

Another widely studied plant-derived polyphenolic derivative, second only to RES in terms of pharmacological importance and beneficial health effects, is curcumin (CUR; Figure 3) [73], a bright yellow-orange diarylethene-structured secondary metabolite found in the tuberous root (rhizome) of various turmeric species, particularly that of *Curcuma longa* L. [74]. The spectrum of pharmacological activities and pharmacokinetic limitations of CUR roughly overlaps that of RES [75], and the mechanisms of action (not fully elucidated) by which it exerts therapeutic effects on HCC cells are mainly the induction of apoptosis, inhibition of tumor cell invasion and metastasis, immunomodulation, and scavenging of free radicals [76,77,78].

Huang M. et al. used CUR as a bioactive component to fabricate a nanosystem consisting of a poly(ethylene glycol) methyl ether *block* poly(lactide-*co*-glycolide (PEG-PLGA) core structure coated with galactosylated chitosan (GCS) [79]. PLGA is a hydrophobic copolymer well-known for its biocompatibility, biodegradability, and non-toxicity and has also been approved by the FDA for systemic administration [80]. PEG (also approved by the FDA for formulations for human use) is a non-ionic hydrophilic polyether employed in this case as a hydrophilic modifier component of PLGA [81]. PEG-PLGA NPs were coated with GCS with the aim of obtaining a targeting delivery system as chitosan (CS) modified with galactose (G) is specifically recognized and internalized (via endocytosis) by asialoglycoprotein receptors (ASGPRs), which are cell surface glycoproteins highly expressed in liver tumor tissues [82]. The conjugate GCS was obtained by means of a standard amide bond formation via NHS-mediated chemistry and using EDCI as a coupling reagent between lactobionic acid (4-*O*-β-galactopyranosyl-d-gluconic acid) and CS (Figure 3). Then, a double emulsion–solvent evaporation approach was employed to afford the NPs assembly with functionalized CS and loaded with CUR, i.e., CUR-GCS@NPs (CUR-CS@NPs were also obtained for comparative studies). This surface-functionalized nano-formulation (HD = ~ 109 nm with a regular spherical shape, EE of ~94%, DL = 4.56%) showed a favorable biosafety profile (up to 500 µg/mL), efficient targeting properties (receptor-mediated internalization by HepG2 cells assessed both in vitro and in vivo plus accumulation within xenograft tumor tissue through enhanced permeability and retention effects) and stable slow DR capability. The antiproliferative activity against HepG2 cells was improved only to a minor extent compared to free CUR and CUR-CS@NPs, i.e., CUR-GCS@NPs → IC_50_ = 8.9 µg/mL vs. CUR-CS@NPs → IC_50_ = 10.8 µg/mL vs. CUR → IC_50_ = 12.0 µg/mL [79].

Anter H.M. et al. used the same nanosystem for hepatocyte attachment (NPs coated with GCS) and preparation technique (double emulsion–solvent evaporation method) to obtain PLGA-based GCS@NPs loaded with apocynin (APO) in place of CUR as a phytochemical component [83]. APO (4′-hydroxy-3′-methoxyacetophenone, also known as acetovanillone; Figure 4) is a cell-permeable phenolic derivative, discovered in the roots of *Apocynum cannabinum* and *Picrorhiza kurroa*, that acts as a potent, reversible and selective inhibitor of NADPH oxidase [84]. Because NADPH oxidase is an enzyme that reduces O_2_ to O_2_•^−^, APO is effective in preventing the formation of superoxide and thus combating OS and inflammation [85]. 

The morphological analyses carried out on APO-GCS@NPs showed spherical architecture with a smooth surface and a characteristic halo-like appearance traceable to the GCS coat of the NPs’ surface. Physical characterization studies showed an EE of ~34%, hydrodynamic diameter (HD) of ~224–232 nm (stable over six months), and a pH-dependent DR (31–60% after 72 h) supposedly governed by PLGA degradation. Noteworthy, the antiproliferative activity against HepG2 cells was increased ~5.6-fold compared to free APO (i.e., IC_50_ = 25.7 µg/mL vs. IC_50_ = 143.2 µg/mL, for APO-GCS@NPs and APO, respectively) [83]. 

A similar targeting strategy via ASGPRs was exploited by Huang Y. et al. to develop CUR-loaded NPs made up of bovine serum albumin (BSA) as a base component [86]. In this case, the galactosyl (G) targeting units were joined to BSA by reductive amination of 4-*O*-β-d-galactopyranosyl-d-glucopyranose using sodium cyanoborohydride. The base conjugate G-BSA was then transformed into CUR-loaded NPs by a desolvation method (i.e., slow addition of ethanol solution of CUR into aqueous solution of G-BSA under vigorous stirring) and using glutaraldehyde as a cross-linking agent to afford CUR-G-BSA-NPs with a spherical shape (~ 116 nm) and relatively narrow size distribution. In vitro studies evidenced a high DR rate and improved CUR bioavailability. EE and DL were set at ~55% and ~14%, respectively. Biological assessments performed on HepG2 cells confirmed effective CUR-G-BSA-NPs receptor-mediated uptake, significant cytotoxicity (IC_50_ = 46 µg/L), induction of apoptosis via NF-κB-p65-mediated mechanism, wound healing properties, and inhibition of cell migration [86].

Instead, Zhao X. and co-workers sought to exploit the ability of CUR to exert a synergistic effect with Doxorubicin (Dox), one of the most used chemotherapeutic agents with limited effects in the management of HCC [87], by preparing lipid NPs loaded with both compounds using a high-pressure microfluidics technique [88]. These Dox/CUR-NPs (spherical in shape with a smooth surface and uniform particle size of ~89 nm; high EE assessed as 97.1% and 99.8% for Dox and CUR, respectively, together with sustained release profile) were comparatively studied in diethylnitrosamine(DEN)-induced HCC in mice together with CUR-NPs, Dox-NPs, and blank lipid NPs after weekly intravenous administration. A significant reduction in liver/body weight ratio, hepatic nodule size, serum hepatotoxicity marker levels (ALT and AST), proliferation and angiogenesis marker levels (C-myc, PCNA, and VEGF), and increased expression of proapoptotic markers (caspase-3 and Bax/Bcl-2 ratio) were observed for the co-delivery NPs compared to the others, therefore supporting the Dox/CUR synergistic effect in vivo. Cell-based assays also supported the Dox/CUR synergistic effect. Interestingly, the increase in cytotoxic effect by Dox/CUR-NPs towards free Dox was more pronounced against the drug-resistant strain (Bel-7402/5-FU cells; IC_50_ = 0.31 µg/mL vs. IC_50_ = 28.33 µg/mL) than against the drug-sensitive strain (Bel-7402 cells; IC_50_ = 0.15 µg/mL vs. IC_50_ = 0.61 µg/mL), indicating that CUR might reverse multidrug resistance through these pathways [88]. 

Barbinta-Patrascu M.-E. and co-workers used a turmeric aqueous extract for the green synthesis of Ag/AgCl-NPs, which were in turn loaded into biomimetic membranes functionalized with a labeling agent (i.e., chlorophyll *a*) and a coating agent (i.e., CS) [89]. The resulting biohybrids underwent various biological assessments, including the antiproliferative effects on HepG2 cells. Turmeric extract, rich in curcuminoids including CUR, demethoxy-CUR, and bisdemethoxy-CUR, acted as a reducing agent to obtain the Ag-based NPs from AgNO_3_ providing at the same time the bioactive components. The functionalized biomimetic membranes were prepared by hydration of soybean thin film forming vesicular structures (liposomes). The eco-design of the final biohybrids entailed a bottom-up approach consisting of a strong stirring of the two components followed by ultrasound treatment in an ultrasonic bath. Biohybrids devoid of individual components were synthesized for comparative studies. Exhaustive morphological and structural analyses confirmed that all these biohybrids were in the nanoscale dimensions (~ 45 nm) and that Ag/AgCl-NPs were adsorbed on the biomimicking lipid bilayer. They also displayed good antioxidant activities (76–93%), hemocompatibility, and antiproliferative effects on HepG2 cells, in particular, the complete nanosystem (IC_50_ = 27.7 mg/mL) [89]. 

A green approach was also exploited by Andelwahab T.S. et al. to obtain FA-targeted AgNPs containing caffeic acid (CA; Figure 5) [90]. CA (3,4-hydroxycinnamic acid) is a very abundant phenolic acid, found in both free and functionalized forms (mainly esters with (-)-quinic acid or sugars, but also amides, glycosides, or more complex structures such as dimers or trimers) in the plant kingdom (e.g., coffee beans, fruits, olives), that has been greatly associated with beneficial effects toward HCC [91]. The main pharmacokinetic issue with CA relies on the fact that human tissues and biological fluids lack specific enzymes to release it from the complex forms in which it is contained in foods and natural products. 

In this research work, FA, upon activation of its terminal -COOH group as NHS ester using DCC as a coupling reagent, was chemically joined to CA (realistically through one of the phenolic moieties) and the resulting conjugate FA-CA was employed both as a reducing and capping agent for the formation of AgNPs. The final nanosystem FA-CA@AgNPs (particle size 10–20 nm) showed good antiproliferative activity against HepG2 cells (IC_50_ = 7.7 µM) and the ability to induce apoptosis (affecting caspase-8, caspase-3, and TNF-α pathways) with or without IR radiation exposure, although the latter was evaluated in vivo in a mouse model of Erlich solid carcinoma [90].

Among the compounds of plant origin that are of greatest interest because of their marked antioxidant properties are undoubtedly flavonoids. Particularly (but not exclusively) investigated in the field of medical-pharmaceutical research are their potential anticancer properties and the preventive/beneficial role that these compounds are able to exert in the presence of diseases of different origins and nature [92]. Among these, quercetin (Q; 3,3′,4′,5-7-pentahydroxy-flavove; Figure 6) is one of the most studied, both for the multiplicity of its pharmacological properties, including anti-inflammatory, immune-stimulating, risk-reducing cardiovascular diseases and metabolic disorders, and for its abundance in nature (berries, grapes, onions, citrus fruits, apples) where it is found more often as a glycoside of various sugars [93]. 

Bishayee K. et al. (2015) used Q as a reducing agent for the synthesis of AuNPs (starting from gold chloride), which were in turn capsulated with PLGA with the help of a stabilizer (1% polyoxyethylene-polyoxypropylene; F68) and conjugated at their surface to a fluorescent probe for biological assessments on HepG2 cells [94]. The latter indicated that Q-AuNPs (spherical shape with a smooth surface; diameter of ~114 nm) undergo considerable cellular uptake and accumulation within mitochondria and nuclei leading to cell cycle arrest at the sub-G stage and apoptosis via the p53-ROS crosstalk pathway. Moreover, circular dichroism measurements showed the Q-AuNPs cause conformational changes in DNA and modulate proteins via epigenetic modifications such as downregulation of HDAC-Akt activities. More importantly, the cytotoxicity assays showed that Q-AuNPs preferentially kill cancer cells (LD_50_ = 24 µg/mL) compared to normal WRL-68 cells (LD_50_ = 75 µg/mL) [94]. Two years later (2017), Ren K.-W. and co-workers reproduced the exact same nano-formulation (spherical shape with a smooth surface; diameter of ~107 nm) for more in-depth biological studies both in vitro and in vivo by means of four human liver cancer lines (i.e., MHCC97H, Hep3B, HCCLM3, and Bel-7402) and a xenograft mouse model (i.e., MHCC97H cells injected subcutaneously into BALB/c nu/nu nude mice), respectively [95]. From these new studies emerged that Q-AuNPs suppress liver cancer progression and development affecting different signaling pathways, including the inactivation of caspase/cytochrome *c* pathway, inhibition of telomerase reverse transcriptase (hTERT) through AP-2β expression, inhibition of COX-2 expression via suppression of the NF-κB nuclear translocation and its binding to COX-2 promoter, and suppression of the Akt/ERK1/2 pathway [95]. 

One of the forms in which Q is often found in nature is rutin (RU; Figure 7), also named quercetin-3-*O*-rutinoside, rutoside, or sophorin, that is, Q with the hydroxy group at position C-3 substituted with glucose and rhamnose. In the glycoside form, Q (as well as all flavonoids) is more soluble in water and therefore more bioavailable than the aglycone form, so it is also preferred for use in the preparation of nano-formulations [96].

Pandey P. et al. used RU as a bioactive component for the development of RU-loaded PLGA-based NPs obtained by double emulsion–solvent evaporation method [97]. The most interesting outcomes for this nano-formulation came from the in vivo preclinical and biochemical studies, performed against DEN-induced HCC in rats, in which it displayed significant improvements in hepatic and hematological parameters after oral administration, including remarkable reduction of OS and levels of inflammatory markers, an increase in the expression of antioxidant enzymes (e.g., GPx, GTS, MPO, CAT, and SOD), downregulation of inflammatory mediators (i.e., IL-1β, IL-6, TNF-α, and NF-κB), and improvement of membrane-bound enzyme activity (Ca^2+^-ATPase, Na^+^/K^+^-ATPase, and Mg^2+^-ATPase) responsible for the destruction of liver tissue. Moreover, histopathological analysis of the liver tissue evidenced a reduction in hepatic nodules and necrosis formation, infiltration of inflammatory cells, blood vessel inflammation, and reduction of liver damage markers such as ALT, AST, γ-glutamyl transferase (GGT), and ALP. RU-PLGA-NPs showed particle size of ~211 nm, DL = 6.39%, EE = 77.83%, and in vitro gastric stability in media with various pHs, along with a biphasic pattern of DR (initial burst release at 4 h, then sustained release over 48 h with a maximal DR of 71%) [97].

A flavonoid with high structural similarity to Q is hesperetin (HP; 3′5,7-trihydroxy-4methoxy flavanone, Figure 8), which is abundant in citrus fruits. This potent antioxidant was loaded on the surface of AuNPs synthesized from Au salts using the polymer *O*-[2-(3-mercaptopropionylamino)ethyl]-*O*’-methyl polyethylene glycol (mPEG_5000_-SH) as both a reducing and a stabilizing agent [98]. The terminal –SH group of the polymer was the actual reducing function, which also served to bind Au to the surface of the AuNPs. The latter were eventually capped with HP. This nano-formulation developed by Gokuladhas K. et al., namely HP-mPEG_5000_-S-AuNPs, underwent the similar biological assessments above-described for RU-PLGA-NPs, including in vivo preclinical and biochemical studies in DEN-induced HCC in rats, determination of the level of all inflammatory markers, antioxidant enzymes and ATPase activity related to liver damage along with the histological analysis of liver tissues in terms of number and size of nodules and tumor tissue growth. All these parameters were ameliorated by treatment with HP-mPEG_5000_-S-AuNPs to a significantly greater extent than by treatment with free HP [98]. 

A phenolic derivative with remarkable antioxidant properties, still little known in the Western world and in pharmaceutical research, is honokiol (HK; Figure 9). It is a biphenolic compound isolated from the root, bark, and leaves of the *Magnolia* species, plants that have been widely used in traditional Chinese and Japanese medicine for a very long time. Recent studies highlighted the anticancer potential of HK both in vitro and in vivo [99,100]. However, despite the low molecular weight, HK suffers from the same pharmacokinetic issues as its parent plant-derived phenolic derivatives, limiting its application as a therapeutic agent.

In order to overcome such issues, Tang P. et al. designed and synthesized a nano-drug delivery system in which HK was loaded into a polymer backbone, i.e., chitin (CH), functionalized with a second bioactive component, i.e., epigallocatechin-3-gallate (EGCG; Figure 10) [101]. CH is the most abundant naturally occurring biopolymer after cellulose. It is a major component of the exoskeleton of insects and other arthropods and the cell wall of fungi and is present in the surface structures of many other invertebrates. Chemically, it is a polysaccharide consisting of *N*-acetyl-glucosamine units (specifically poly(β-(1→4)-*N*-acetyl-d-glucosamine) [102]. 

The synthetic strategy entailed the initial deacetylation (50% NaOH) of CH to obtain free -NH_2_ groups, which were in turn protected with phthalic anhydride. Then, phthaloyl-CH was functionalized with EGCG (to afford phthaloyl-CH-EGCG or phthaloyl-CE) by a grafting procedure consisting of a free radical reaction between the functional groups of the two components. An H_2_O_2_/ascorbic acid redox pair was employed to generate the free radicals. Eventually, the de-phthaloylation of the conjugate was performed using hydrazine monohydrate. The final CE-HK-NPs were prepared by an ionic cross-linking method. This nano-formulation turned out to be stable (~80 nm particles with spherical shape) and non-toxic, providing a sustained release of the bioactive component in vitro (80% after 24 h). Its antitumor activity was evaluated both in vitro (HepG2 cells) and in vivo (HepG2 tumor-bearing mice). The in vitro studies showed that CE-HK-NPs exert antiproliferative activity in the G2/M phase and decrease mitochondrial membrane potential. The in vivo studies evidenced a remarkable reduction in tumor growth (~84%) compared to free HK (~30%) after intertumoral injection thrice a week over a 15-day treatment [101].

In the field of natural (poly)phenolic substances with antioxidant activity incorporated into nano-formulations for the treatment of HCC, the work of Kumar V. et al. unifies in a way all the features described so far both in terms of rational design, preparation/characterization techniques of the nano-platform for the delivery of the bioactive compound, and in vitro and in vivo experimental models suitable for the validation of the results. The nanosystem in this case is obtained by a double emulsion–solvent evaporation method consisting of a PLGA polymer matrix loaded with a coumarin derivative in the form of a glycoside, namely umbelliferone β-d-galactopyranoside (UFG; Figure 11), whose galactose sugar functions as an active targeting element for ASGPRs [103]. UF was selected as a bioactive component because it is well-known that it exhibits (besides the antioxidant activity) antiproliferative activity in HCC cells via induction of apoptosis and cell cycle arrest [104]. 

UFG-PLGA-NPs displayed uniform size distribution (~187 nm), EE = 60–90%, and sustained in vitro DR (82.5% after 48 h). The in vivo assessments were performed on DEN-treated rats, one of the most accepted experimental models for hepatocarcinogenesis, in which UFG-PLGA-NPs were able to reduce the liver/body weight ratio and the number of liver nodules together with a wide panel of standard parameters of liver damage, including AST, ALP, and ALP serum levels, ROS generation, expression of pro-inflammatory cytokines, and mitochondrial dysfunction. In vitro studies revealed that UFG-PLGA-NPs inhibit cell proliferation of Huh-7 and HepG2 cells in a dose-dependent manner [103]. 

## 3. Nano-Formulations Containing Terpenoid Antioxidants

Terpenoids are another interesting class of naturally occurring compounds endowed with a wide range of pharmacological activities, including antioxidant, anti-inflammatory, immunomodulatory, and anticancer activities [105,106]. Most terpenoids are synthesized by plants in the chloroplasts (starting from the lipophilic 5-carbon unit isoprene via the photosynthesis-dependent 2-C-methyl-d-erythritol-4-phosphate pathway) toward which they mainly play a protective role against thermal and oxidative stress [107]. However, terpenoids can also be synthesized by bacteria and yeasts as part of primary and secondary metabolism. The antioxidant action of these compounds is due to the isoprene unit itself, which being endowed with a C=C unsaturation reacts and quenches ROS [108]. Based on the number of the isoprene building blocks, terpenoids are classified into monoterpenes (2 units), sesquiterpenes (3), diterpenes (4), triterpenes (6), tetraterpenes (8), and polyterpenes (> 8). From recent studies (including preclinical animal models), they have emerged as a promising group of phytochemicals for the chemopreventive and therapeutic treatment of HCC, especially in view of the fact that they selectively kill liver cancer cells with a pleiotropic mode of action while sparing normal cells [109]. 

One of the most effective phytochemicals of the terpenoid class is andrographolide (AG; Figure 12), a major diterpene lactone isolated from *Andrographis paniculata* (Kalmegh). This plant is typical of the traditional system of Chinese and Indian medicine, where it is suggested as hepato-protective and hepato-stimulative with antioxidant effects against various hepatotoxins, and whose activities have been related to this bioactive terpenoid [110]. 

Das S. et al. used AG to prepare PLGA-based NPs loaded with this phytochemical ingredient that turned out to be ~5-fold more effective than free AG against arsenic (supplied via drinking water as NaAsO_2_ at a dose of 40 mg/L for 30 days) induced liver damage in mice after oral administration on alternate days [111]. AG-PLGA-NPs were obtained by emulsion–solvent evaporation method and showed an average particle size of ~66 nm, EE = 64%, and in vitro DR of 50% on day 8 and 100% on day 20. This nano-formulation remarkably decreased serum levels of liver damage markers (i.e., ALT, AST, and ALP) as well as arsenic deposition in the liver. In addition, it decreased the level of hepatic antioxidant enzymes such as SOD and CAT, and the level of GSH [111].

Khan M.W. and co-workers used oleanolic acid (OA) as a bioactive component to develop a pH-dependent stimuli-responsive nanosystem co-loaded with the anticancer drug cisplatin (CDDP) [112]. OA ((3β)-3-hydroxy-olean-12-en-28-oic acid; Figure 13) is the most abundant pentacyclic triterpenoid in the plant kingdom widely known for its hepatoprotective, free radical scavenging, and antitumor properties [113,114]. 

Previous studies have demonstrated that OA is able to attenuate Dox-induced multi-organ toxicity in HCC when co-delivered in liposomal formulation [115]. In Khan’s study, OA was shown to alleviate CDDP-induced hepatotoxicity, assessed by determination of ALT and AST levels and histochemical analysis, while exerting a synergistic effect with the chemotherapeutic agent, assessed by cell cycle analysis and apoptosis studies. In this nano-formulation, CDDP was loaded as a prodrug (i.e., nitrate salt) into calcium carbonate (CC) cores through a water-in-oil microemulsion method. Specifically, two microemulsions were obtained before mixing, breaking, and centrifugation: calcium emulsion (from CaCl_2_ and oil phase CO-520) and carbonate emulsion (from Na_2_CO_3_ and oil phase CO-520); CDDP was added to the aqueous phase of the carbonate emulsion. Then, the CDDP-CC cores underwent an outer lipid coating containing OA through the emulsion–solvent evaporation method. The final CDDP/OA-LCC@NPs had a spherical shape (~206 nm), EE = ~64%, and a pH-dependent in vitro DR (i.e., 70% of CDDP at pH = 5.5 and 28% of CDDP at pH = 7.4). The biological assessments revealed that CDDP/OA-LCC@NPs induce apoptosis via downregulation of the P13K/Akt/mTOR pathway and upregulation of the p53 proapoptotic pathway, which is in accordance with the anticancer mechanisms previously determined for OA [116]. Moreover, the co-presence of OA and CDDP in the nano-formulation clearly helped CDDP to overcome the drug-resistance issue by downregulating proteins like XIAP and Bcl-2 via the NK-κB pathway [112].

OA was also used as a bioactive component by Bao X. et al. to prepare OA-loaded NPs composed of PLGA-d-α-tocopheryl PEG_1000_ succinate (PLGA-TPGS) random copolymer as a biodegradable matrix [117]. TPGS is a versatile water-soluble PEG-derivative of vitamin E that finds application in drug delivery research as an excellent emulsifier, solubilizer, and bioavailability enhancer of hydrophobic drugs through multiple mechanisms, including increasing drug permeability across the cell membrane and inhibiting the multidrug resistance transporter P-glycoprotein [118,119]. These OA-PLGA-TPGS-NPs, obtained by the ultrasonic emulsion–solvent evaporation method, showed a spherical shape (~200 nm), DL = ~28%, EE = ~92%, and a biphasic DR pattern (initial burst followed by a sustained release). Biological studies evidenced efficient cellular uptake, remarkably higher cytotoxicity (IC_50_ = 0.9 µg/mL after 72 h) against HepG2 cells compared to free OA (IC_50_ = 48.9 µg/mL after 72 h) as well as compared to NPs lacking the TPGS (IC_50_ = 7.3 µg/mL) component and reference anticancer drug 5-FU (IC_50_ = 18.3 µg/mL), and better tumor-inhibiting effects in vivo (ascitic HCC strain HCa-F inoculated into mice) compared to the same standards (growth inhibition rate in volume = 53.6%) [117].

An uncommon terpenoid with very special characteristics is parthenolide (PLT; Figure 14), a sesquiterpene lactone contained in the medicinal plant feverfew (*Tanacetum parthenium*), which is used as an inhibitor of the nuclear transcription factor NF-κB [120]. Its structure contains two highly reactive functional groups, namely an epoxide ring and a lactone ring, which are susceptible to nucleophilic attack by biomolecules such as DNA and proteins. Feverfew as a herbal product is widely used for its therapeutic properties, including the anti-inflammatory property presumably attributable to this terpenoid. However, the lack of water solubility and poor bioavailability limit the use of PLT as a drug. 

This terpenoid was recently employed as a bioactive component together with tyrosol (TYR) by Baharani et al. for the development of hybrid NPs coated with CS and lecithin (L) [121]. TYR (2-(4-hydroxyphenyl)-ethanol) is a natural water-soluble phenolic compound found in several plant species, particularly in olive oil, in the form of esters with fatty acids. Although it is not as potent as other natural antioxidants, its higher chemical stability and bioavailability than other poly(phenols) indicate that it may have an important overall effect in protecting cells from injury due to OS [122]. The final hybrid nanosystem PLT/TYR-CSL@NPs (average size of ~38 nm; EE = 93% of PLT) was obtained by an auto-self-assembling method consisting of dropwise addition of the lipid phase solution (ethanol 96%) containing L and PLT to the aqueous phase solution (acetic acid 1%) containing CS and TYR. It showed cancer-selective cytotoxicity in vitro assessed on HepG2 cells (IC_50_ = 95 µM) and normal fibroblast HFF cells (no significant inhibition), potent antioxidant activity (IC_50_ = 187 µg/mL and IC_50_ = 290 µg/mL in ABTS and DPPH test, respectively), and substantial apoptotic effects by upregulating the expression of the apoptotic genes Bax and caspase-8 and downregulating the expression of the anti-apoptotic gene Bcl-2 [121]. 

## 4. Nano-Formulations Containing Supplemental Antioxidants

Besides plant-derived antioxidants, there are numerous other natural antioxidants with similar therapeutic and/or preventive potential for managing liver cancer that can be exploited in nanomedicine, including, for example, certain vitamins such as vitamin C (Vit. C) and vitamin E (Vit. E). Vit. C is one of the most potent antioxidants in humans, but it is not synthesized due to the lack of the enzyme gluconolactone oxidase [123]. It is a water-soluble lactone with six carbon atoms that exists in nature in two enantiomeric forms, of which only one (the enantiomer (5*R*)-5-[(1*S*)-1,2-dihydroxyethyl]-3,4-dihydroxyfuran-2(5H)-one; Figure 15) is considered Vit C. Vit. E (Figure 15), on the other hand, is lipid-soluble and consists of a group of eight tocopherols, of which α-tocopherol is the most active form and the one considered Vit. E in the absence of further specification. It also exerts potent antioxidant activity and has a potent function in improving the immune system, stress, and disease resistance [124]. 

Aljuhr S.A. et al. used these two vitamins as bioactive components for the coating of selenium NPs [125]. The rationale for the design of this nanosystem relies upon the fact that selenium is another powerful antioxidant component and that in the form of SeNPs, it is much more bioavailable, biocompatible, and biodegradable in vivo than the other forms in which it occurs in the diet, namely selenite (SeO_3_^−2^), selenate (SeO_4_^−2^), and organo-selenium compounds [126,127]. The Vit. E/C@SeNPs were obtained through a two-step process. First, Vit. C was used in an aqueous medium as both a reducing agent for Na_2_SeO_3_ and a coating agent for the resulting SeNPs. Second, the resulting Vit. C@SeNPs were further coated with Vit. E, which was dissolved in acetonitrile and added to the mixture. This nano-formulation showed efficacy in vivo in DEN/CCl_4_-induced HCC in rats with considerable reduction in liver function biomarkers (ALT, AST, ALP, total bilirubin, and GGT), increase in GSH concentration and CAT activity, together with marked improvements in the histological features of liver tissue. qPCR analyses also revealed significant upregulation of the expression of various inflammatory and apoptotic genes. Vit. E/C@SeNPs (average core size of ~50 nm) also showed high antioxidant properties (~76% of radical scavenging capacity; DPPH test) and acceptable cytotoxicity (IC_50_ = 28 µg/mL assessed in normal MRC-5 cells by sulforhodamine B method) [125].

A different Vit. E component, specifically α-tocotrienol (Figure 16), was used by Tupal A. and co-workers for the development of solid lipid NPs (SLNPs) containing Precirol^®^ ATO5 (glyceryl distearate) as a base component for the incorporation of this supplemental antioxidant [128]. 

Precirol^®^ ATO5 is a high lipophilic glyceride that can be used as a powerful platform for the controlled release of bioactive agents from dosage forms. SLNPs formed with this component have also shown high tumor tissue accumulation and biocompatibility [129]. These tumor-targeted SLNPs were obtained by means of a hot high-shear homogenization method that entailed the solubilization of the bioactive compound in a liquid oil (miglyol) and its addition to the base component with the support of a proper surfactant (poloxamer 407). They turned out to be effective at enhancing the cytotoxicity of Dox as well as the cytotoxicity of free α-tocotrienol against Huh-7 HCC cells detected by apoptotic studies (18% → 37% and 21% → 33% of apoptotic cells, respectively) and cell cycle studies (arrest of HCC cells in sub-G1 phase up to 66%). These SLNPs showed an average particle size of ~78 nm, efficient cellular internalization, and ability to increase the cytotoxicity of α-tocotrienol in vitro (IC_50_ = 15 µM vs. IC_50_ = 10 µM). Furthermore, RT-PCR studies revealed a significant decrease in the expression of anti-apoptotic genes such as survivin and Bcl-2, and an increase in proapoptotic genes, including Bid and Bax [128].

A powerful natural antioxidant that can act directly as a ROS scavenger and indirectly regenerate the cellular Vit. C and Vit. E, discussed earlier, is coenzyme Q10 (CoQ10, also known as ubiquinone; Figure 17). This endogenous lipid-soluble and essential vitamin-like substance exerts several protective functions within the human body, including that of proton–electron transporter in the mitochondrial respiratory chain resulting in energy production for the cell, protection of membrane lipids from peroxidation, hepatoprotection from OS, and xenobiotic-associated cellular damage. In spite of this huge protective potential, CoQ10 is endowed with poor bioavailability that limits its clinical applications especially if orally administered [130]. 

The protective effects of CoQ10 against hepatotoxicity and mitochondrial/lysosomal injury induced by the organophosphate pesticide dichlorvos in rats were evidenced by Eftekhari A. et al. through a nano-formulation in which CoQ10 was formulated by the precipitation method as naked nanocrystals and coarse suspensions [131]. The resulting CoQ10-NPs (mono-dispersed and with an average diameter of ~54 nm) underwent exhaustive biological studies to evaluate fundamental parameters for maintenance of normal liver function, including ROS level, lipid peroxidation, cell viability, enzymatic liver damage markers, antioxidant enzyme activities, lysosome membrane integrity, cellular GSH content, and mitochondrial membrane potential. Overall, all these parameters were improved by nano-formulation compared with CoQ10 in non-particulate form, particularly the ROS formation, lipid peroxidation, and cytotoxicity [131].

In the work by Quagliarello V. et al., CoQ10 was used to achieve both cardioprotection and hepatoprotection in human cardiomyocytes and hepatocytes during exposure to anthracyclines and human epidermal growth factor (HER2) inhibitors, in particular to Dox and monoclonal antibody Trastuzumab, respectively [132]. CoQ10 was first encapsulated in oil-in-water nano-emulsions (o/w NEs) in order to increase its oral bioavailability. Then, these o/w NEs were coated with a layer of CS that may provide selective biodistribution in cardiomyocytes and hepatocytes [133]. Then, CoQ10-CS@NPs were coated with a layer of hyaluronic acid (HA) with the aim of obtaining an active-targeting delivery nanosystem as HA recognizes CD44 receptors that are overexpressed in heart and liver tissue [134]. CoQ10-CS/HA@NPs had HD = ~ 122 nm that was relatively stable over six weeks, high loading ability, and increased cell viability (protection rate 47–53%) in both cell lines during treatments with Dox and Trastuzumab. Furthermore, under the same anticancer treatment conditions, this nano-formulation turned out to be effective at inhibiting lipid peroxidation and inflammation via reduction of the expression of leukotriene B4, p65/NF-κB, and pro-inflammatory cytokines such as IL-1β and IL-6 [132].

Docosahexaenoic acid (DHA; Figure 18) is a natural omega-3 polyunsaturated fatty acid that has been found to possess remarkable antioxidant and anticancer properties against several malignant neoplasms [135,136]. This highly lipophilic compound represents an important component of the human brain, cerebral cortex, retina, and skin. DHA shows an affinity for LDL. Therefore, the latter can be envisaged as suitable nanocarriers for the delivery of this bioactive compound, which can be internalized into cells through endocytosis by LDL-specific receptors [137]. 

Moss L.R. and co-workers developed DHA-loaded LDL NPs that turned out to be selectively cytotoxic to liver cancer cells (i.e., TIB-75 → IC_50_ = 27.7 µM DHA) over normal hepatocytes (i.e., TIB-73 → IC_50_ = 91.4 µM DHA) and primary hepatocytes (IC_50_ = 148.8 µM DHA) by downregulating ROS, causing lipid peroxidation and lysosome leaking, and promoting mitochondrial and nuclear damage. These NPs (average size of ~20 nm) were obtained by the reconstitution (core-loading) method, namely extraction with heptane of lyophilized LDL and addition of DHA to the LDL residue [138]. All the discussed nano-formulations, along with their physical characteristics and biological activity, are listed in Table 1. 

In the context of nano-formulations associated with the management of OS and the treatment of HCC, special mention should be made of trace elements (TEs), which are those elements in ionic form that are required in very minute quantities for the regular functioning and development of organisms [139]. An impaired balance of these TEs has been associated with the development of various types of cancers, including HCC [140]. The TEs most involved in cellular homeostasis are iron, copper, zinc, and selenium [141,142]. Very schematically, it can be stated that carcinogenesis is associated with excessive levels of iron and copper and deficient levels of zinc and selenium, although the issue is more complex, and experimental results are not always so linear with what has been stated.

Iron is involved in several cellular processes such as DNA synthesis, the mitochondrial respiratory chain, energy metabolism, and oxygen transport through integration with proteins containing heme as a prosthetic group. In relation to the development of HCC, the most accepted hypothesis is the generation of ROS by excess free Fe^2+^ unbound to plasma transferrin or cytoplasmic ferritin [143]. In the past decade, numerous nanosystems have been developed for the treatment of HCC containing iron in the form of iron oxide (Fe_3_O_4_) NPs obtained mainly by the co-precipitation or thermolysis method. These nanosystems, although inherently nonmagnetic, have superparamagnetic properties and thus have the ability to be magnetized with the application of an extrinsic magnetic field. These properties can be exploited in nanomedicine for magnetic resonance (MR) imaging, for magnet-mediated drug delivery, and for inducing hyperthermia using heat as a therapeutic for liver cancer. Most relevant and recent achievements in this regard include the development of (i) SPION-loaded lysosomes that can be actuated with a pulsed magnetic field treatment inducing apoptosis in HepG2 and Huh-7 cells [144]; (ii) nanocomposites for passive targeting delivery of siRNA in hepatic tumor-bearing mice [145]; (iii) hydrophilic and surface-functionalized SPIONs for treating liver cancer via magnetic fluid hyperthermia [146]; (iv) lipid-coated Fe_3_O_4_ NPs targeting Huh-7 cells via ASGPRs for dual-modal fluorescence and MR imaging [147]; (v) delivery nanoplatforms loaded with the multi-targeted kinase inhibitor sorafenib to enhance its antitumor effectiveness against HepG2 cells [148]; (vi) Dox-loaded and FA-functionalized delivery nanoplatforms with enhanced bioavailability and efficacy against Hep3B cells [149]; (vii) glucose-coated Fe_3_O_4_ NPs loaded with coumarin with significant antiproliferative activity against HepG2 cells compared to normal hepatocyte cells THLE2 [150]; (viii) CUR-capped Fe_3_O_4_ NPs with remarkable activity against HepG2 cells [151]; (ix) delivery nanosystems for active liver tumor-targeting imaging (through GA receptors) with good biocompatibility and ultralow hepatoxicity both in vitro (HepG2 and Huh-7 cells) and in vivo (mouse models bearing subcutaneous and orthotopic liver tumors) [152]; (x) Fe_3_O_4_ NPs coated with glucose and conjugated with safranal (a natural and potent monoterpenic antioxidant isolated from saffron) showing efficient antiproliferative activity against HepG2 cells through cell cycle arrest and apoptosis induction [153].

Copper is another TE essential for a wide range of physiological processes involving defense against OS. In the form of extracellular Cu^2+^, it is reduced to Cu^1+^ by transmembrane proteins in the reductase family (STEAP proteins) before being absorbed. Excess Cu^1+^ can be toxic or even lethal to cells, as it causes ROS production and subsequent cellular damage. However, copper deficiency can also be harmful to the human body, as it impairs the function of several enzymes and the immune system. Therefore, copper homeostasis is regulated by sophisticated mechanisms that regulate its cellular uptake (through the high-affinity membrane protein for Cu^1+^ CTR1), efflux (copper-transporting P-type enzymes ATP7A/B that shuttle between the Golgi apparatus and the cell membrane) intracellular compartmentalization (metallothioneins) and buffering (copper chaperones CCS and ATOX1 that transport Cu^1+^ from the cytosol to the SOD1 enzyme and ATP7A/B transporters, respectively) [154]. There is increasing evidence linking elevated serum copper levels to the development of HCC and the accumulation of copper in the liver tissue of HCC patients [155].

Copper-based nano-formulations consist of NPs of copper (II) oxide (CuO) preferentially obtained by an environmentally friendly approach (green synthesis), although other methods have been employed, including electrochemical reduction, chemical synthesis, and microwave irradiation. The green method has proven to be the preferred one mainly because CuO NPs (and metal-based NPs in general) obtained by chemical procedures suffer from the adsorption of hazardous chemicals on their surface, which limits their biomedical applications. In addition, the green method is non-toxic, cost-effective, and provides NPs that are more stable, homogeneous, and of adequate size [156]. Recent research works pertaining to the potential use of CuO NPs for the treatment of liver cancer include (i) green synthesis of CuO NPs using leaf extracts of *Azadirachta indica* as a reducing agent for Cu(II) salt (i.e., copper(II) acetate monohydrate) with activity against HepG2 cells [157]; (ii) green synthesis of CuO NPs using leaf extracts of *Momordica cochinchinensis* (Lour.) as a reducing agent for copper(II) sulfate with activity against HepG2 cells [158]; (iii) chemical synthesis (from copper(II) acetate monohydrate and NaOH solution) of CuO NPs capable of inducing cytotoxicity and mitochondria-mediated apoptosis in HepG2 cells [159].

Zinc is also a TE that plays a crucial role in numerous biochemical pathways, being a cofactor in more than 300 mammalian proteins, many of which are enzymes involved in antioxidant defense [160]. Zinc levels are regulated by two types of transporters: The ZnT family, which actuates zinc efflux from cells and sequesters zinc from intracellular organelles, and the ZIP family, which is responsible for zinc uptake from extracellular fluid [161]. Dysregulation of zinc homeostasis has been associated with the development of HCC [162]. Specifically, zinc deficiency reduces the expression of metallothioneins, cysteine-rich proteins to which zinc binds and which mediate detoxification and protection against OS. As for copper, the oxide form (ZnO) is the preferred one in terms of biocompatibility for the preparation of nano-formulations. Relevant research works in this regard include (i) chemical and green synthesis (seed extracts of *Nigella sativa* and zinc nitrate) of ZnO NPs with significant antiproliferative activity against HepG2 cells [163]; (ii) green synthesis of ZnO NPs using leaf extracts of *Aquilegia pubiflora* that are highly toxic to HepG2 cells via ROS production [164]; (iii) chemical and biogenic synthesis (plant extract of *Celosia argentea* and zinc acetate) of ZnO NPs that induced apoptosis in HepG2 cells via OS [165]; (iv) green synthesis of ZnO NPs using leaf extracts of *Moringa oleifera* (rich in (poly)phenolic components) both as a reducing and a capping agent. These NPs showed hepatoprotective effects in CCl_4_-treated albino rats [166]; (v) synthesis of ZnO NPs with efficacy on HCC induced in rats treated sequentially with DEN and CCl_4_ [167]; (vi) development of ZnO NPs capable of reducing serum levels of HCC markers (α-fetoprotein and α-l-fucosidase), as well as hepatocyte integrity and OS markers in DEN-treated rats [168]; (vii) ZnO NPs fabricated using the crustacean immune molecule β-1,3- glucan binding protein as a coating agent that showed cytotoxicity against HepG2 cells [169].

Selenium is an important TE that exerts antioxidant and detoxifying actions for the human body by acting primarily as a cofactor of enzymes (selenoproteins) that include GPx and TrxR. However, the optimal amount of selenium for regular cell function is within a very narrow range. Therefore, in this case, it is also more appropriate to debate selenium homeostasis. Its deficiency facilitates ROS production due to decreased activity of selenoproteins; an excess, on the other hand, is toxic to the body due to substantial oxidation of thiol groups [170]. In relation to the occurrence of HCC, the inverse correlation between selenium levels and the incidence of this type of malignancy appears clear, although some studies have yielded controversial results, leaving the question open [171]. In this context, SeNPs (whose extraordinary properties have been discussed above) may exert both antioxidant and pro-oxidative activities, increasing the expression of selenoproteins (hepatoprotective properties) and causing ROS production (and consequent activation of ROS-mediated apoptosis signaling pathways) in cancer cells to a greater extent than in normal cells, respectively, because of their exclusively hepatic metabolism [172,173].

Extensive studies dealing with SeNPs for the treatment of HCC have been carried out in recent years. These studies include (i) G-modified SeNPs for active targeted delivery of Dox in HepG2 cells [174]; (ii) FA-targeted and polyethylenimine(PEI)-decorated SeNPs as delivery vehicles for siRNA [175]; (iii) multifunctional SeNPs with baicalin (a flavone glycoside found in several plant species of the genus *Scutellatia*) and FA for the treatment of HBV-infected HepG2215 cells [176]; (iv) CUR-surface decorated SeNPs capable of inducing HepG2 cell apoptosis through ROS-mediated p53 and Akt signaling pathways [177]; (v) sorafenib-doped SeNPs with increased cytotoxicity against HepG2 cells [178]; (vi) siRNA-loaded and surface-PEI-decorated SeNPs modified with HA (for active tumor-targeting) showing relevant anticancer activity in vitro (HepG2 cells) and in vivo (xenograft mouse model) [179]; (vii) multifunctional SeNPs loaded with the antioxidant flavonol galangin that induces apoptosis in HepG2 cells through p38 and Akt signaling pathways [180]; (viii) PEI-decorated SeNPs as carries of HSP70 siRNA to induce apoptosis in HepG2 cells [181]; (ix) green synthesis of SeNPs obtained from hawthorn fruit extracts with significant cytotoxic activity against HepG2 cells [182]; (x) SeNPs stabilized by β-glucan nanotubes from the fruiting body of black fungus with remarkable cytotoxicity against HepG2 cells [183].

## 5. Challenges and Future Perspectives

The nanomedicine–natural substance combination is the new frontier toward which all major pharmaceutical companies are moving. Nanomedicine, namely the medical application of the possibilities offered by nanotechnology, offers enormous potential in terms of the delivery of bioactive compounds that can be administered in the most appropriate formulation and site to achieve the greatest possible efficacy while containing the risks of adverse reactions. Depending on the matrix characteristics of NPs (from lipids in SLNPs to functionalizable organic polymers to real metals), it is possible not only to control the solubility and release time of a bioactive compound but also to obtain a stimuli-responsive delivery platform and/or effect a multiphase therapeutic treatment through the release of a single active ingredient in multiple stages or by allowing the simultaneous administration of two or more active compounds that can subsequently be released at different times from each other, going on to bind their specific targets with synergistic effects. These features prove to be crucial in the field of oncology (both in terms of therapy and early diagnosis through the identification of tumor biomarkers), where selectivity of action is essential. The use of natural substances, for its part, intersects well with the medicine of the future, which is increasingly geared toward preventing and curing disease through natural stimulation of the body’s functions and enhancement of its defense systems, and with the green approach in the use of natural resources and methods of synthesizing bioactive compounds in terms of eco-sustainability. Although the potential of nano-formulations of natural antioxidants for liver cancer treatment is known, several challenges remain and must be addressed to fully take advantage of their therapeutic benefits. One of the main challenges is the need for extensive clinical trials. While preclinical studies have shown promising results, there is a substantial gap between laboratory success and clinical applicability. Clinical trials are crucial to establish the safety, efficacy, and long-term effects of these nano-formulations in human patients with different liver functions, genetic factors, and disease stages. Another critical challenge lies in the manufacturing process. The production of nano-formulations on a large scale presents a significant barrier, particularly in ensuring batch-to-batch consistency, stability, and quality control. Therefore, optimizing the design of nano-formulations by using a continuous manufacturing process will help to control reproducibly the size, shape, and surface characteristics of such nanoparticles, thereby enhancing the targeting specificity and reducing the potential side effects. Additionally, the combination of nano-formulated antioxidants with other therapeutic treatments (e.g., chemotherapy, immunotherapy, radiotherapy) is interesting research to pursue to support the standard of care in liver cancer. The future of antioxidant nano-formulations for liver cancer treatment is promising but requires deeper investigations. In conclusion, while challenges remain, the potential of nano-formulations of natural antioxidants in liver cancer treatment is undeniable. With continued research and innovation, these challenges can be addressed, paving the way for more effective, targeted, and personalized therapies that may significantly improve outcomes for patients with liver cancer.

## Figures and Tables

**Figure 1 biomolecules-14-01031-f001:**
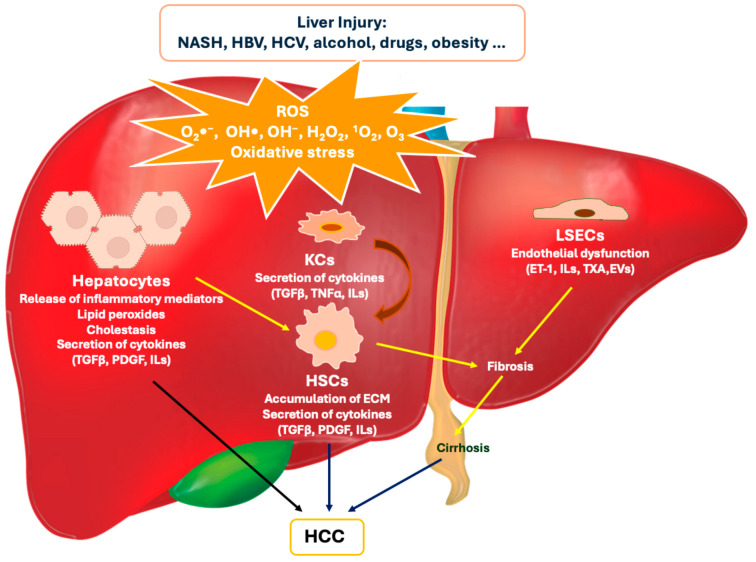
Schematic representation of the molecular mechanisms and cascade pathways involved in liver damage and related to oxidative stress (OS). Intercellular communication that occurs during liver injury includes sinusoidal endothelial cells (LSECs) capillarization, Kupffer cells (KCs) activation, and hepatic stellate cells (HSCs) activation. These activated cells release warning signals, including pro-inflammatory cytokines and growth factors, which contribute to the progression and persistence of liver damage. HBV: hepatitis B virus; HCV: hepatitis C virus; NASH: nonalcoholic steatohepatitis; HCC: hepatocellular carcinoma; ECM: extracellular matrix; ET-1: endothelin-1; EVs: extracellular vesicles; ILs: interleukins; PDGF: platelet-derived growth factor; TGFβ: transforming growth factor β; TNFα: tumor necrosis factor α; TXA: thromboxane.

**Figure 2 biomolecules-14-01031-f002:**
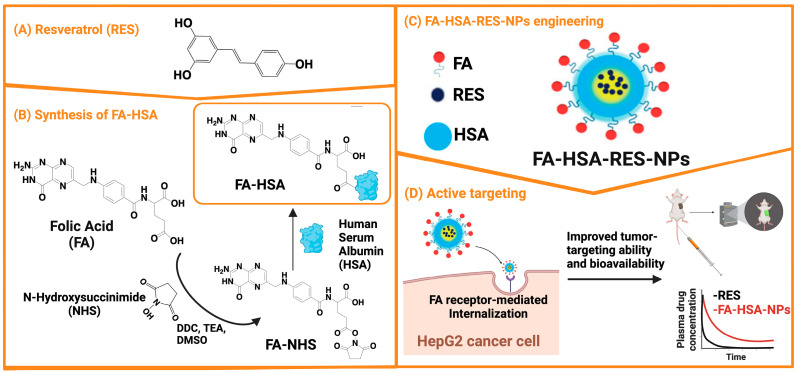
Schematic representation of the targeted drug delivery system developed by Lian B. et al.: (**A**) Chemical structure of RES; (**B**) Synthesis of the FA-HSA conjugate; (**C**) Representation of the FA-HSA-RES-NPs obtained by the encapsulation of RES within FA-HSA using a high-pressure fluid nano-homogeneous emulsification method; (**D**) Proposed mechanism of active tumor targeting.

**Figure 3 biomolecules-14-01031-f003:**
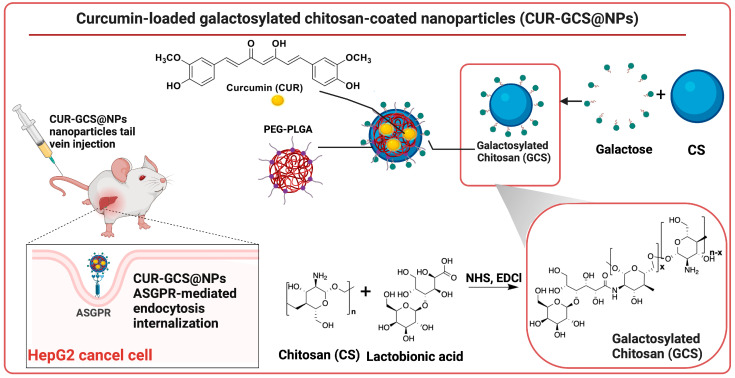
Schematic representation of the synthesis of galactosylated chitosan (GCS) and its application in curcumin-loaded nanoparticles (CUR-GCS@NPs) developed by Huang M. et al. These GCS-coated NPs were designed to target the asialoglycoprotein receptors (ASGPRs) overexpressed on hepatocellular carcinoma cells, offering enhanced biocompatibility and potential therapeutic efficacy.

**Figure 4 biomolecules-14-01031-f004:**
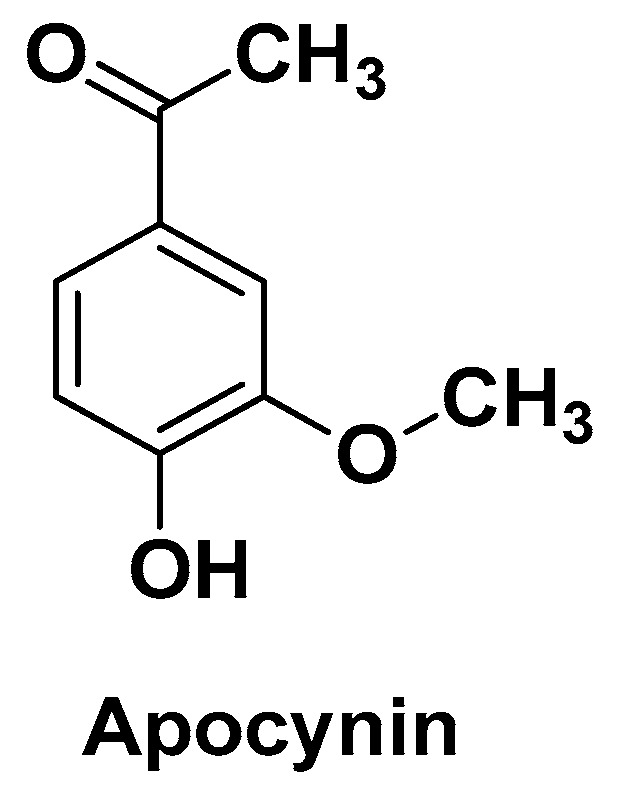
Chemical structure of apocynin (APO).

**Figure 5 biomolecules-14-01031-f005:**
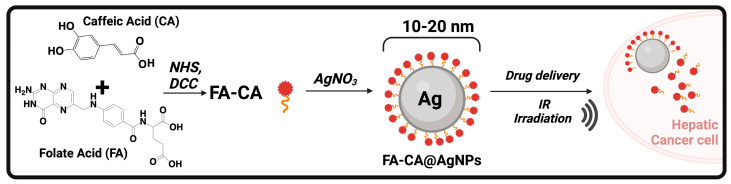
Schematic representation of the tumor-targeting and IR-sensitive nanosystem (FA-CA@AgNPs) designed and developed by Abdelwahab T. et al. Caffeic acid (CA) was used both to chemically conjugate the targeting agent (folic acid; FA) and as a reducing agent to form AgNPs.

**Figure 6 biomolecules-14-01031-f006:**
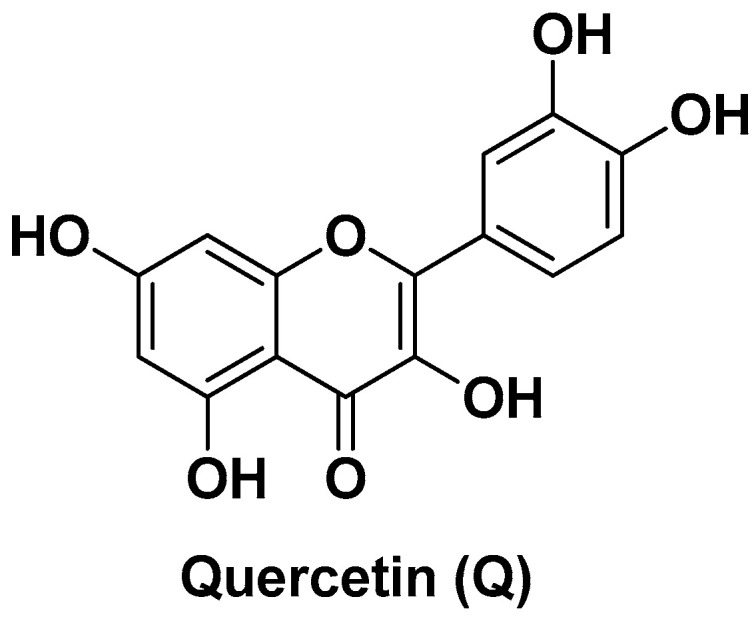
Chemical structure of quercetin (Q).

**Figure 7 biomolecules-14-01031-f007:**
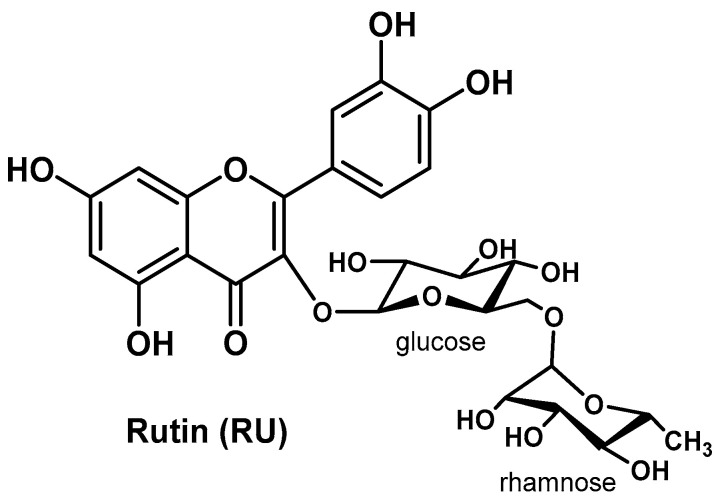
Chemical structure of rutin (RU).

**Figure 8 biomolecules-14-01031-f008:**
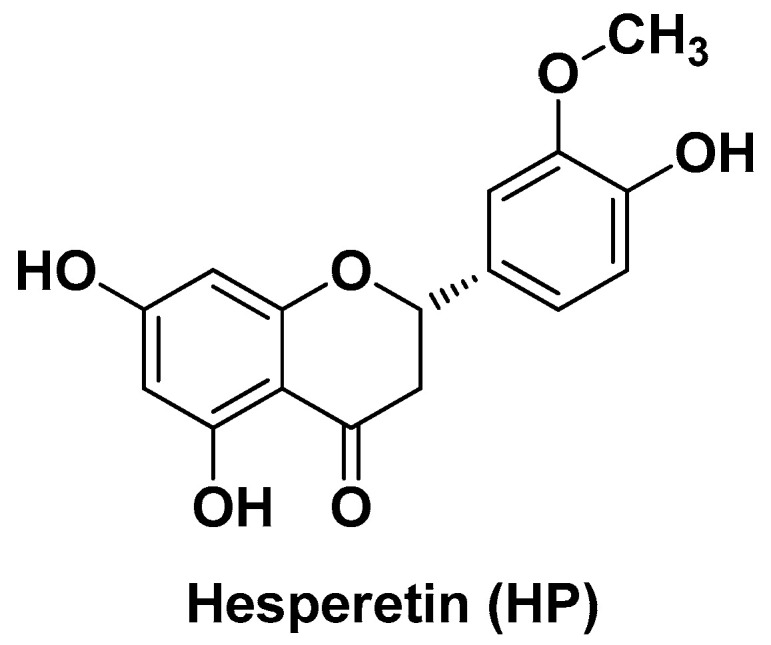
Chemical structure of hesperetin (HP).

**Figure 9 biomolecules-14-01031-f009:**
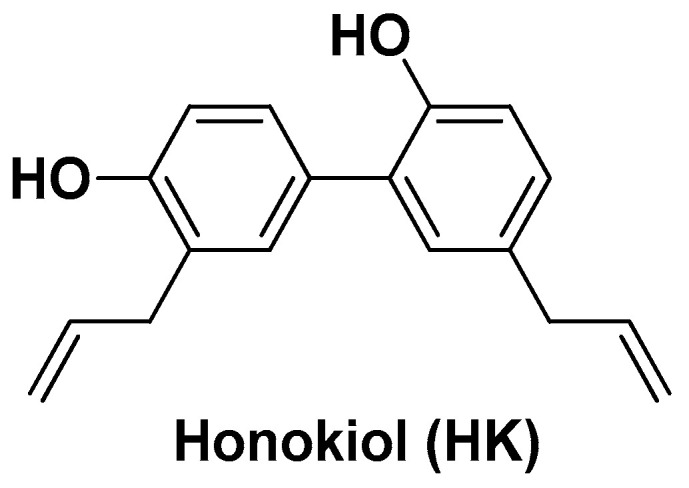
Chemical structure of honokiol (HK).

**Figure 10 biomolecules-14-01031-f010:**
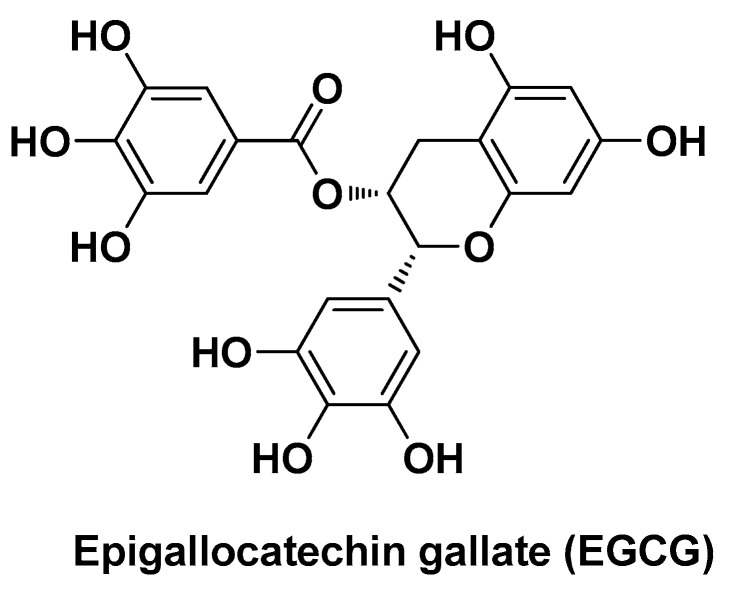
Chemical structure of epigallocatechin gallate (EGCG).

**Figure 11 biomolecules-14-01031-f011:**
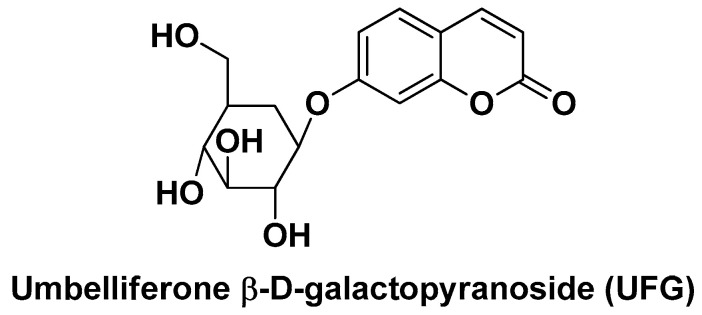
Chemical structure of umbelliferone β-d-galactopyranoside (UFG).

**Figure 12 biomolecules-14-01031-f012:**
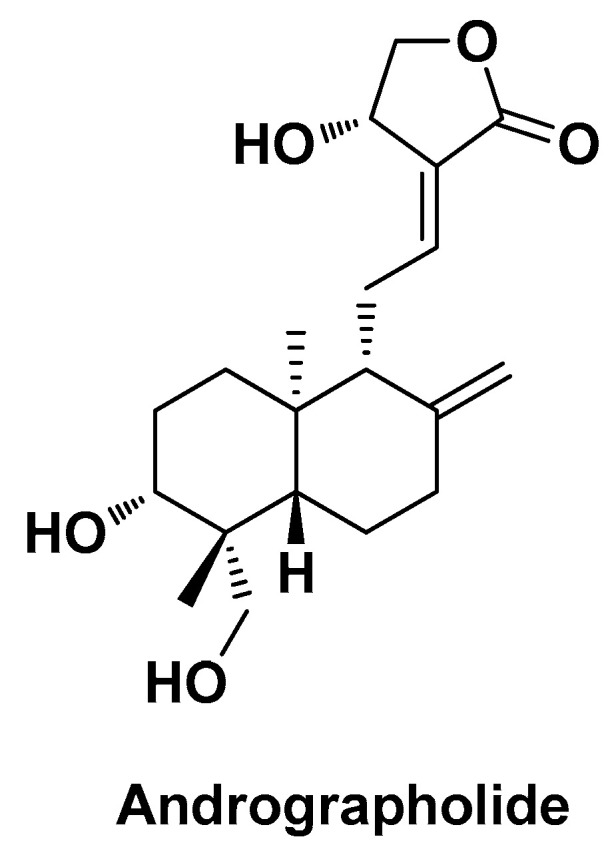
Chemical structure of andrographolide (AG).

**Figure 13 biomolecules-14-01031-f013:**
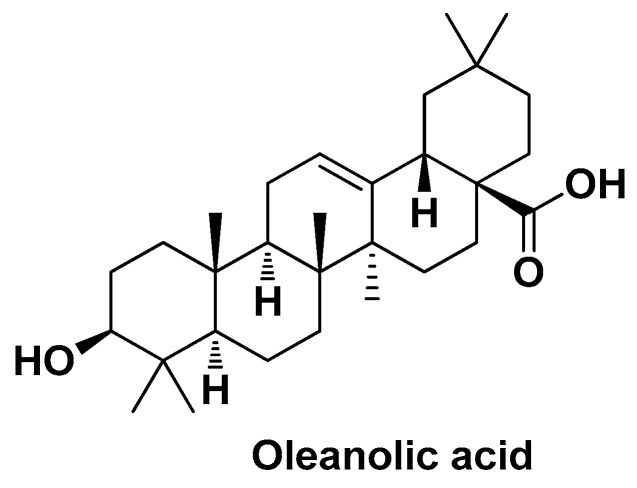
Chemical structure of oleanolic acid (OA).

**Figure 14 biomolecules-14-01031-f014:**
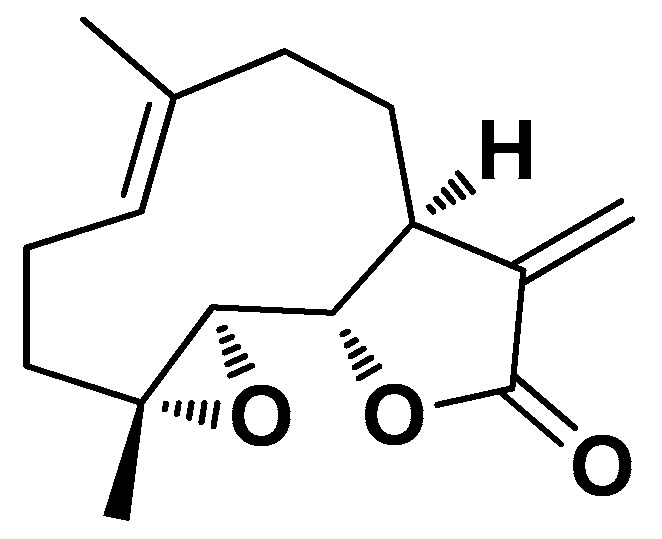
Chemical structure of parthenolide (PLT).

**Figure 15 biomolecules-14-01031-f015:**
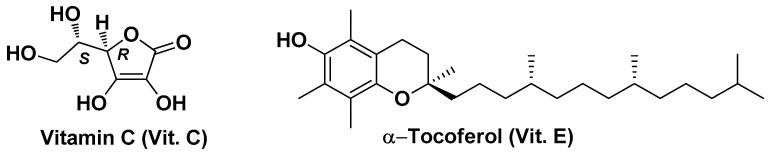
Chemical structure of vitamin C (Vit. C) and vitamin E (Vit. E).

**Figure 16 biomolecules-14-01031-f016:**
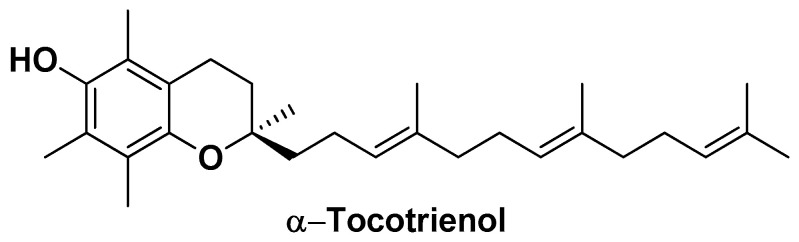
Chemical structure of α-tocotrienol.

**Figure 17 biomolecules-14-01031-f017:**
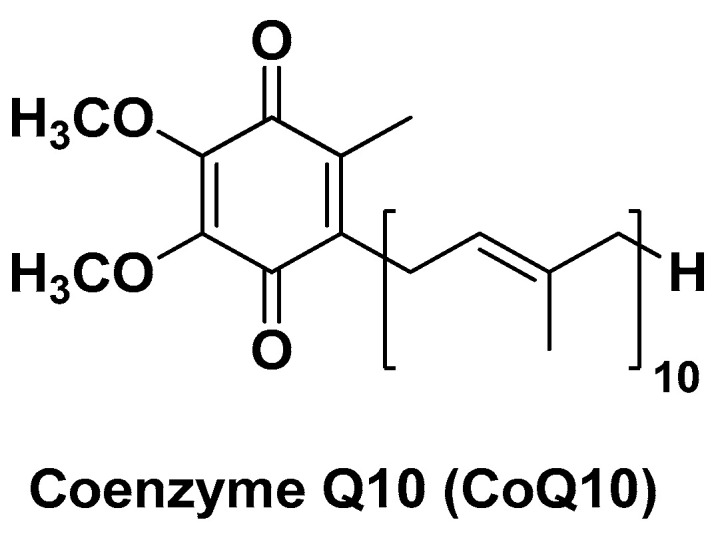
Chemical structure of coenzyme Q10 (CoQ10).

**Figure 18 biomolecules-14-01031-f018:**
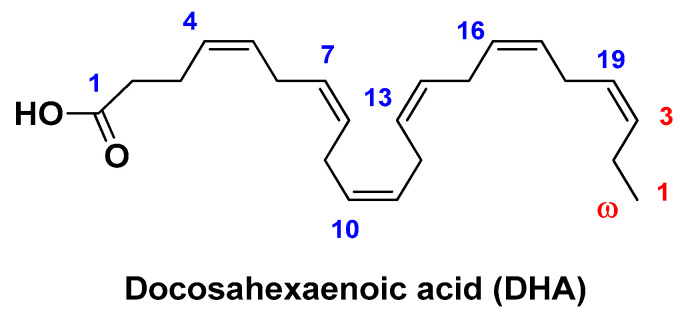
Chemical structure of docosahexaenoic acid (DHA).

**Table 1 biomolecules-14-01031-t001:** Nano-formulation’s characteristics and in vitro and/or in vivo biological activity in liver cancer.

Nano-Formulation	API	Excipients	Physical Characteristics	Biological Activity	Ref.
FA-HSA-RES-NPs	Resveratrol	Folic acid, human serum albumin	Spherical shape (~102 nm), high EE (~98%), and DL (~15%)	Enhanced cell uptake and antiproliferative activity in vitro and improved biodistribution	[64]
GA-HSA-RES-NPs	Resveratrol	Glycyrrhizic acid, human serum albumin	Nearly spherical (~108 nm), high EE (83.6%), and DL (11.5%)	Enhanced antiproliferative activity in HepG2 cells and better biodistribution via EPR effect and ASGPR-mediated endocytosis	[65]
RES-AuNPs	Resveratrol	Gold nanoparticles	Spherical shape (~39–1110 nm) and zeta potential (ζ) = −32.5 mV	Stronger apoptosis induction via downregulation of pro-caspase-9, pro-caspase-8, PI3K, and Akt and upregulation of caspase-8 and Bax, tumor growth suppression.	[70]
CUR-GCS@NPs	Curcumin	PLGA-PEG, galactosylated chitosan	Particle size ~100 nm, EE = ~94%, DL = 4.56%, ζ = −9.82 mV, Polydispersity index (PDI) = 0.25	Targeted delivery to liver tumor tissues, enhanced biocompatibility	[79]
APO-GCS@NPs	Apocynin	PEG-PLGA, galactosylated chitosan	Particle size 224–232 nm, EE = 34%, pH-dependent drug release (31–60% after 72 h)	Significant improvement of antiproliferative activity against HepG2 cells	[83]
CUR-G-BSA-NPs	Curcumin	bovine serum albumin, galactosyl units	Spherical shape (~116 nm), high drug release rate	Better antiproliferative activity against HepG2 cells (~5,6-fold compared to the free drug)	[86]
Dox/CUR-NPs	Doxorubicin, Curcumin	Glyceril distearate, soybean lecithin, Polyoxyl 40 Hydrogenated Castor Oil, glycerin, triglycerides medium chain	Spherical shape (~89 nm), high EE (Dox = 97.1% and CUR = 99.8%), ζ = −14.3 mV, PDI = 0.22, sustained release profile (Dox = 55% and CUR = 40% after 48 h)	Synergistic antiproliferative effects in HCC models, modulation on apoptosis, proliferation-, angiogenesis-, MDR- and hypoxia-related mRNAs and proteins	[88]
FA-CA@AgNPs	Folic acid, Caffeic Acid	Silver nanoparticles	Small particle size (10–20 nm)	Higher antiproliferative effects, apoptosis induction by caspase-8, caspase-3, and TNF-α pathways	[90]
Q-AuNPs	Quercetin	Gold nanoparticles	Spherical shape (~114 nm)	Higher cellular uptake and apoptosis induction via the p53-ROS pathway	[94]
RU-PLGA-NPs	Rutin	PLGA	Particle size ~211 nm, high EE (77.83%) and DL (6.39%), sustained release (71% after 48 h)	Overexpression of GPx, GTS, MPO, CAT, and SOD, downregulation of IL-1β, IL-6, TNF-α, and NF-κB, and improvement of membrane-bound enzyme activity (Ca^2+^-ATPase, Na^+^/K^+^-ATPase, and Mg^2+^-ATPase)	[97]
HP-mPEG_5000_-S-AuNPs	Hesperetin	mPEG_5000_-SH, gold nanoparticles	Spherical, triangular, and pentagon in shape (110–120 nm), ζ = −4.38 mV, sustained release (80% after 72 h)	High antioxidant activity, improved hepatic parameters in HCC models, inhibition of inflammatory markers, antioxidant enzymes, and ATPase activity related to liver damage	[98].
CE-HK-NPs	Honokiol	Chitin, epigallocatechin-3- gallate	Spherical shape (~80 nm), sustained release (80% after 24 h)	Extended antiproliferative activity in vitro and reduction of tumor growth (~84%) after inter-tumoral injection (3x week)	[101]
UFG-PLGA-NPs	Umbelliferone β-d-galactopyranoside	PLGA	Uniform size distribution (~187 nm), EE = 60–90%, sustained in vitro DR (82.5% after 48 h)	Reduced liver/body weight ratio and liver nodules in DEN-treated rats, inhibition of HCC cell proliferation in vitro.	[103]
AG-PLGA-NPs	Andrographolide	PLGA	Particle size ~66 nm, EE = 64%	Decreased serum levels of ALT, AST, and ALP, arsenic deposition in the liver, SOD, CAT, and GSH	[111]
CDDP/OA-LCC@NPs	Oleanolic Acid, Cisplatin	Mono-methoxy polyethylene glycol 2000-distearoyl phosphatidylethanolamine (PEG-DSPE 2000) 1,2-dioleoyl-in-glycerol-3-phosphate (DOPA), dehydrogenated soya phosphatidylcholine (HSPC), Calcium carbonate	Particle size ~206 nm, EE = ~64%, pH-dependent drug release (70% of CDDP at pH = 5.5 and 28% of CDDP at pH = 7.4)	Induction of apoptosis via downregulation of P13K/Akt/mTOR pathway and upregulation of p53 proapoptotic pathway, inhibition of drug resistance by downregulating proteins like XIAP and Bcl-2 via the NK-κB pathway	[116]
OA-PLGA-TPGS-NPs	Oleanolic Acid	PLGA, d-α-tocopheryl PEG_1000_ succinate	Spherical shape (~200 nm), DL = ~28%, EE = ~92%	Increased in vitro cytotoxicity against HepG2 cells compared to the free drug, and higher growth inhibition rate in volume	[117]
PLT/TYR-CSL@NPs	Parthenolide, Tyrosol	Chitosan, Lecithin	Particle size ~38 nm; PLT EE = 93%	Cancer-selective cytotoxicity in vitro assessed on HepG2 cells and potent antioxidant activity. Apoptotic effects by upregulating the expression of the apoptotic genes Bax and caspase-8 and downregulating the expression of the anti-apoptotic gene Bcl-2	[121]
Vit. E/C@SeNPs	Vitamin C, Vitamin E	Selenium nanoparticles	Particle size ~50 nm, high antioxidant capacity (~76% DPPH scavenging), improved liver function markers	Higher antioxidant capacity (~76% DPPH scavenging), improved ALT, AST, ALP, total bilirubin, and GGT, increased GSH concentration and CAT activity	[125]
Precirol^®^ ATO5	α-Tocotrienol	Precirol ATO5 Glyceryl distearate, miglyol, poloxamer 407	Particle size ~78 nm, ζ = −11 mV, PDI = 0.24	Increased cytotoxicity in vitro (IC_50_ = 15 µM compared to IC_50_ = 10 µM of the free drug), decreased expression of the anti-apoptotic genes survivin and Bcl-2, and increased expression of the proapoptotic genes Bid and Bax	[128]
CoQ10-CS/HA@NPs	Coenzyme Q10	Hyaluronic acid, Chitosan	Monodispersed, average diameter ~54 nm	Hepatoprotective effects against OS and xenobiotics, enhanced cellular antioxidant capacity	[131]
DHA-LDL-NPs	Docosahexaenoic acid	LDL	Particle size ~20 nm	Selective cytotoxicity against liver cancer cells, modulation of oxidative stress and mitochondrial damage	[138]

## Data Availability

Not applicable.

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
