# Peer review of "Nano-Formulations of Natural Antioxidants for the Treatment of Liver Cancer"

_biomolecules, 2024, doi:10.3390/biom14081031_

Round 1

Reviewer 1 Report

Comments and Suggestions for Authors

Dear authors,

the present manuscript is an interesting review about nano-formulations of natural antioxidants, which are an emerging therapeutic approach in the treatment of liver cancer.

Please highlight in the abstract and introduction section the novelty of this review and why now. The abstract has to be reconsidered as

The benefits of nano-formulations in Liver Cancer Treatment not well highlighted. Please elaborate 2-3 sentences for each benefit, e.g. enhanced bioavailability, targeted delivery, control release, synergic effect, etc...

Nano-formulations can encapsulate bioactive compounds, protecting them from degradation, improving their solubility, and enabling targeted delivery to tumor sites. Please refer to this aspect when describing the compounds and their activity.

I suggest as well to ass a new section, Challenges and Future Perspectives, as nano-formulations of natural antioxidants show great promise, but several challenges remain, e.g. extensive clinical trials are necessary to validate the effectiveness of nano-formulations of natural antioxidants in treating liver cancer.

Comments on the Quality of English Language

minor spelling

Author Response

Reviewer #1

Comments and Suggestions for Authors

Dear authors,

the present manuscript is an interesting review about nano-formulations of natural antioxidants, which are an emerging therapeutic approach in the treatment of liver cancer.

  • We thank the reviewer for his/her valuable and positive feedback on our manuscript. We appreciate his/her thoughtful suggestions and agree that they will significantly enhance the clarity and impact of our work.

Please highlight in the abstract and introduction section the novelty of this review and why now. The abstract has to be reconsidered as the benefits of nano-formulations in Liver Cancer Treatment not well highlighted. Please elaborate 2-3 sentences for each benefit, e.g. enhanced bioavailability, targeted delivery, control release, synergic effect, etc... Nano-formulations can encapsulate bioactive compounds, protecting them from degradation, improving their solubility, and enabling targeted delivery to tumor sites. Please refer to this aspect when describing the compounds and their activity.

  • We revised the abstract and introduction to clearly highlight the novelty of this review as requested, emphasizing the main benefits and the relevance of the topic at the present time. However, due to the word limit in the abstract (200 words maximum), we were unable to include 2-3 sentences for each benefit as suggested by the reviewer but we did our best to accommodate the request.

I suggest as well to add a new section, Challenges and Future Perspectives, as nano-formulations of natural antioxidants show great promise, but several challenges remain, e.g. extensive clinical trials are necessary to validate the effectiveness of nano-formulations of natural antioxidants in treating liver cancer.

  • We appreciated this Reviewer's suggestion. We have added a new section titled "Challenges and Future Perspectives," which addresses the remaining challenges in the development of nano-formulations of natural antioxidants. This section discusses the necessity for extensive clinical trials, highlights other key challenges, and outlines potential future research directions. This section has been partially merged with the previous “Conclusion” section.

Reviewer 2 Report

Comments and Suggestions for Authors

This paper reviewed the most important and recent (last decade) scientific findings  of novel nano-formulations containing natural antioxidant molecules targeted for the treatment of liver cancer. It is known that natural antioxidants trap free radicals by converting them into non- harmful products and minimize oxidative stress, playing a great activity in the treatment of free radical-induced diseases. However, the most of these antioxidants possess poor bioavailability. Various nanodrugs delivery systems have been developed nowadays to improve their stability, control  release and promote targeted delivery for liver cancer treatment, which is one of the main purpose of human death .  It was analysed and concluding that depending on the characteristics of the nanonoparticles (NPs ) used in nanoformulations from lipid NPs and functionalized organic polymers NPs to metal NPs (silver and gold), it is possible to control the solubility and release-time of a bioactive compound, and to obtain a stimuli-responsive delivery platform with a multiphase therapeutic treatment through the release of a single active ingredient  or by allowing the simultaneous administration of two or more active compounds that can subsequently be released at different times with synergistic effects. 

    The paper is in the scope of Biomolecules and can be published after minor editing in present form.

Comments on the Quality of English Language

The paper is written by good scientific English and can be published after minor editing in present form.

Author Response

Reviewer #2

Comments and Suggestions for Authors

This paper reviewed the most important and recent (last decade) scientific findings of novel nano-formulations containing natural antioxidant molecules targeted for the treatment of liver cancer. It is known that natural antioxidants trap free radicals by converting them into non- harmful products and minimize oxidative stress, playing a great activity in the treatment of free radical-induced diseases. However, the most of these antioxidants possess poor bioavailability. Various nanodrugs delivery systems have been developed nowadays to improve their stability, control release and promote targeted delivery for liver cancer treatment, which is one of the main purpose of human death. It was analysed and concluding that depending on the characteristics of the nanonoparticles (NPs ) used in nanoformulations from lipid NPs and functionalized organic polymers NPs to metal NPs (silver and gold), it is possible to control the solubility and release-time of a bioactive compound, and to obtain a stimuli-responsive delivery platform with a multiphase therapeutic treatment through the release of a single active ingredient  or by allowing the simultaneous administration of two or more active compounds that can subsequently be released at different times with synergistic effects. 

The paper is in the scope of Biomolecules and can be published after minor editing in present form.

  • We sincerely thank the reviewer for his/ positive feedback on our manuscript. We are very glad about his/her appreciation. The article has been revised for minor editing as indicated.

Reviewer 3 Report

Comments and Suggestions for Authors

See the attachment

Comments on the Quality of English Language

need to improve

Author Response

Reviewer #3

Comments and Suggestions for Authors

Overall the present review is very interesting and worthy of the readers of this journal. After my careful evaluation the review need major revision.

  • We thank the Reviewer for having appreciated our article and proving suggestions for further rising its level and interest for the readers.
  1. Author should be discuss trace element metal and metal oxide like ZnO, CuO and selenium based green synthesis nanoparticles and its antioxidant and liver cancer activity should be discuss and make a one table. Many reports already available.
  • It was not easy to accommodate this request without running the risk of straying from the theme of the article, which deals with natural antioxidants in nano-formulations for the treatment of liver cancer. Trace elements are certainly of primary importance in the antioxidant defense of the human body, as they play crucial roles in the management of oxidative stress; in that sense, they may somehow fit into this review. However, they would merit a separate review to cover all the other elements besides the three you mentioned (e.g., iron, silicon, and vanadium), and many reviews actually report them separately. We have done our best to fulfill your request by trying to reward researchers working in this field by citing and briefly reporting recent results related to nano-formulations containing the four most important trace elements (i.e., Fe, Cu, Zu, and Se) that have been designed and tested for the treatment of liver cancer. We decided not to provide a very detailed discussion of each article because there are too many of them (see the 45 citations we had to add) and they clearly require separate review and several months of work. Also, they were not added to the table because mixing nano-formulations based on organic and inorganic active components is somewhat misleading to the reader. We hope this is of for you.
  1. Green nanoformulation antioxidant molecular mechanism should be discuss.
  • Following your preceding request, green syntheses of NPs (mostly related to metal-based nano-formulations) were reported and discussed in the main text, highlighting their importance. Antioxidant mechanisms have also been reported where the authors have given evidence of them.   

Round 2

Reviewer 3 Report

Comments and Suggestions for Authors

Accept

Comments on the Quality of English Language

Minor spell check